# A cost analysis comparing seasonal malaria chemoprevention with and without Vitamin A supplementation among under-5 children in Nigeria

Olusola Bukola Oresanya[1]*, Olujide Arije[2], Jesujuwonlo Fadipe[1], Kunle Rotimi[1], Abimbola Phillips[3], Kolawole Maxwell[4], Emmanuel Shekarau[5], Nneka Onwu[6], Eva S. Bazant[7]

1 Technical, Malaria Consortium, Maitama, Abuja, Nigeria, 2 Institute of Public Health, Obafemi Awolowo University, Ile-ife, Osun, Nigeria, 3 Technical, Former Malaria Consortium, Maitama, Abuja, Nigeria, 4 Managment, Malaria Consortium, Maitama, Abuja, Nigeria, 5 Case management, National Malaria Elimination Programme, Abuja, Nigeria, 6 Community Services, National Primary Health Care Development Agency, Abuja, Nigeria, 7 The Task Force for Global Health, Health Campaign Effectiveness Coalition, Decatur, Georgia, United States of America

* o.oresanya@malariaconsortium.org

## Abstract

### Background

Child mortality in Nigeria, significantly affected by malaria and malnutrition, remains a public health concern in the country. Seasonal Malaria Chemoprevention (SMC) and Vitamin A supplementation (VAS) are effective interventions that can be delivered through integrated health campaigns to reduce this mortality. This study assesses the cost implications of integrating these two interventions among under-5 children in Northeast Nigeria.

### Methods

A cost analysis compared standalone SMC (Cycle 1 in July 2021) with SMC-VAS integrated campaign (Cycle 4 in October 2023) in two Local Government Areas (LGAs) in Bauchi State. The number of children reached by the SMC-only campaign was 168,820 and for the SMC + Vit A campaign, the number was 170,681. Data collection utilized a mixed-methods approach, drawing from primary and secondary sources, including programmatic, financial, and coverage records. Costs were categorized into distribution, Sulphadoxine-Pyrimethamine plus Amodaiquine (SPAQ) for SMC, Vitamin A, training, supplies, meetings, labor, supervision, and social mobilization costs. Sensitivity analyses evaluated the effect of a 10% fluctuation in the costs of distribution, labor, SPAQ, and supplies on the cost per child.

**Data availability statement:** All relevant data are within the paper and its Supporting Information files.

**Funding:** This work was supported, in whole or in part, by the Bill & Melinda Gates Foundation (Grant Number INV-01076 to the Task Force for Global Health's Health Campaign Effectiveness Program). Under the grant conditions of the Foundation, a Creative Commons Attribution 4.0 Generic License has already been assigned to the Author Accepted Manuscript version that might arise from this submission.

**Competing interests:** The authors have declared that no competing interests exist.

## Results

The total cost for the SMC standalone campaign was US$158,934, and the SMC-VAS integration was US$186,426. Distribution and drug costs were the largest contributors in the integrated and SMC-only campaign. The SMC-only cost per child was $0.94 and $1.18 when eligible children received both SMC and VAS. The integration of VAS into the SMC campaign cycle incurred an additional US$27,492 over Cycle 1 cost (US$186,426 – US$158,934). Fluctuations in distribution costs were the most influential component of the cost per child.

## Conclusion

Integrating VAS with SMC campaigns increases the cost by US$0.24 per child, a modest increment considering the potential health benefits. The results support the feasibility of this integration, in terms of cost, to combat child mortality from malaria and malnutrition in Nigeria. Further research is recommended to explore the cost-effectiveness of this integrated distribution model.

## Background

In spite of documented global gains in child survival, one in every thirteen children in sub-Saharan Africa still face huge health risks, especially mortality before their fifth birthday [1]. The leading causes of morbidity and mortality among under-five children are infectious diseases (malaria) and malnutrition with Nigeria being the third largest contributor to under-five mortality [1,2]. Nutrition-related factors heighten the risk of all cause childhood mortality [1,3].

Central to the effort to combat malnutrition among under-five children is Vitamin A supplementation (VAS) as one of the key strategies recommended [4]. It has been linked to significant reductions in morbidity and mortality among children aged 6–59 months across various regions including Africa Asia and Latin America [5]. VAS is cost effective in addressing global public health challenges among children [6,7]. However, VAS coverage in Nigeria stands at 45.3%, below the World Health Organization's (WHO) recommended coverage of 80% [8], with children in northern Nigeria nearly three times less likely to access VAS compared to their counterparts in the South [8]. Moreover, distributing Vitamin A supplements through health facilities often results in unequal access due to several barriers including socio-cultural and economic factors [8,9].

Some countries with low VAS coverage have adopted strategies targeting the distribution of micronutrients, including VAS, through established, culturally acceptable channels like community-based distribution [10,11]. Seasonal malaria chemoprevention (SMC) can provide a ready platform for VAS delivery to enhance coverage and access. This malaria prevention intervention delivers full courses of antimalarials to children living in areas with seasonal transmission of malaria through a door-to-door campaign approach, A feasibility pilot study in northwest

Nigeria which integrated SMC and VAS in 2019 offered preliminary evidence of improved VAS coverage [12]. However, evidence gaps, including feasibility of implementation in different contexts, equity, cost, efficiency, and safety of integration, remain to be addressed before adopting this approach as a strategy. In response to the need to provide additional evidence for policy decision-making on full-scale integration of VAS with SMC campaigns in Nigeria, a comparative cost analysis of a standalone SMC campaign (cycle 1) and an integrated campaign of SMC plus VAS (cycle 4) was carried out.

The objective was to robustly compare the cost per eligible child between a stand-alone SMC campaign cycle and an integrated SMC-VAS campaign cycle, providing valuable insights for estimating the expenses of future interventions aiming to implement VAS, especially in conjunction with SMC.

## Methods

### Study setting

This study was carried out in Bauchi State, where children face considerable health risks. According to the 2018 Nigeria's Nutrition Health Standardized Monitoring and Assessment of Relief and Transition (SMART) Survey report, Bauchi state has the 7th highest rate of malnutrition in Nigeria (25.5%) significantly higher than the national average of 19.9%, ranks 3rd highest with 8.2% prevalence of severe malnutrition, and 2nd highest in North East with stunting prevalence of 45.6% (compared to the national average of 32%) among children 0−59 months [5]. Bauchi State was selected for the integrated campaign because of its low (29%) coverage of Vitamin A among 6−59 months in 2018 according to the National Nutrition and Health Survey, a level lower than the national average of 41% [13]. Giade LGA is rural with 51,839 children under-5 years old in a total population of 259,192 (2021 projected population) while Katagum is urban with 97,743 children under five years in a population of 488,715 (2021 projected population).

### SMC-VAS Integrated Intervention

SMC entails monthly administration of a three-day treatment regimen comprising sulfadoxine-pyrimethamine (SP) and amodiaquine (AQ), collectively referred to as SPAQ, to children aged 3–59 months. Typically, community distributors administer the first dose of SPAQ in person, followed by two additional doses of AQ provided by caregivers on the second and third days following the initial dose. Each complete SPAQ course confers protection for approximately one month. Each treatment sequence, along with its protective duration, is termed an 'SMC cycle' and four to five cycles are typically provided to cover the malaria transmission season. Fig 1 (Typical design of SMC drug distribution campaign) below shows the typical timeline for one cycle of a drug distribution campaign.

Malaria Consortium supported the state malaria programme to implement a four-cycle SMC delivery in ten Local Government Areas (LGAs) in Bauchi State in 2021. SMC-VAS integration implementation research was conducted in two of the ten LGAs, Giade and Katagum, between June and November 2021 because of their location and stable security status in a region that has reported various security threats in the past few years [14].

Inclusion criterion for receiving SPAQ was children 3–59 months of age, and exclusion criteria were: very sick child, child with fever (who was referred to the nearest health facility for a malaria test), child with known allergy to Sulfadoxine pyrimethamine (SP) or Amodiaquine (AQ), or who has taken SP or cotrimoxazole in the last 4 weeks. Inclusion criterion for receiving Vitamin A was children between 6–59 months of age, while children with severe respiratory infection or difficulty with breathing and children who have taken Vitamin A within the past four months were excluded.

SMC drugs and vitamin A were administered to eligible children using community drug distributors (CDDs), who are volunteer members of the communities put forward by their communities to be trained to distribute SPAQs within their communities and they carry out the distribution through a door-to-door delivery strategy during SMC campaigns. Cycles

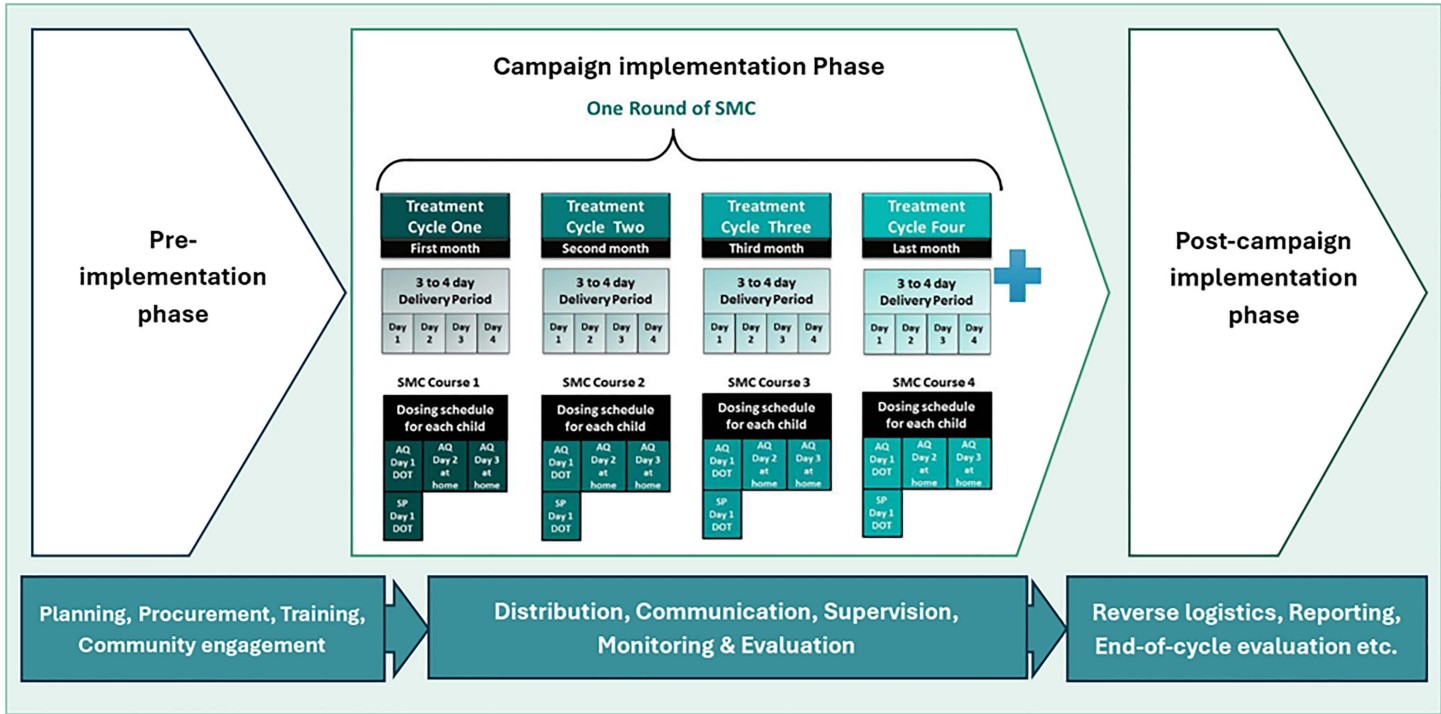

**Fig 1. Typical design of SMC drug distribution campaign.**

1–3 had standalone SMC administration over four days each between June and September 2021, while the integrated delivery of SMC and Vitamin A was implemented during Cycle 4 over five days in October 2021 in the two selected LGAs. The choice of cycle 4, which is the last cycle of SMC, allows for the implementation of SMC-VAS integration without affecting the next SMC cycle within the year.

Eligible children aged six months to less than 12 months received SPAQ1 (Amodiaquine 76.5 mg + Sulphadoxine/Pyrimethamine 250/12.5 mg dispersible tablets) and 100,00 IU Vitamin A, while children who were 12 months to less than 60 months received SPAQ2 (Amodiaquine 153 mg + Sulphadoxine/Pyrimethamine 500/25 mg dispersible tablets) and 200,000 IU Vitamin A (Table 1 and 2). Children who were 3–6months received SPAQ only.

Training of CDDs and supervisors occurred before the onset of Cycle 1, and additional training was held in Cycle 4 for the integration of Vitamin A into the SPAQ campaign. For the SMC-VAS co-delivery, the CDDs followed the standard operating procedure for SMC, with additional steps to guide them on how to determine eligibility, administer Vitamin A to the 6–59-month age group, provide key messages on both interventions and referral of cases. CDDs moved in teams of two persons, one providing treatment of SPAQ and vitamin A, and the other tallying and recording treatments in four records: the SMC Tally Sheet, Growth Monitoring Register, Vitamins A sticker to affix to Child Health Card, and Child SMC Card. CDDs teams were attached to health facilities for commodity logistics and submission of reports. A designated health facility worker provided supportive supervision to the CDD teams attached to the facility. State and LGA teams monitored activities at the facility and community levels using standard checklists while the study teams also monitored the campaign for quality assurance. Ethical approval for the implementation research was provided by the Bauchi State Ethics Committee under the Ministry of Health and written informed consents were received from all participant, who also consented to their data being used for future research so long as they are de-identified. The cost analysis utilized administrative and implementation cost data with no identifiers.

**Table 1. Eligibility criteria for SPAQ and Vitamin A administration.**

| Eligibility for SPAQ (SMC) | Eligibility for Vitamin A supplementation |
|---|---|
| Inclusion criterion | **Inclusion criterion** |
| Between 3–59 months | Between 6–59 months |
| Exclusion criteria | **Exclusion criteria** |
| Child is **5–10 years** | Child is **5–10 years** |
| Child is **very sick** | Child has severe respiratory infection or difficulty with breathing |
| Child has a **fever** | Taken vitamin A in the **past month** |
| Child **allergy** to SP or AQ or cotrimoxazole | Child **allergy** to vitamin A and/or any of its component ingredient |
| Child has taken SP or cotrimoxazole in the last 4 weeks | |

**Table 2. Details of the drugs and supplements used during the campaigns.**

| S/N | Generic Name | Age group |
|---|---|---|
| 1 | Amodiaquine 76.5 mg + Sulphadoxine/Pyrimethamine 250/12.5 mg dispersible tablets (SPAQ1) | Children 3 month to less than 12 months |
| 2 | Amodiaquine 153 mg + Sulphadoxine/Pyrimethamine 500/25 mg dispersible tablets (SPAQ2) | Children 12 months to 59 months |
| 3 | Vitamin A supplement (100,000 IU) | Children 3 month to less than 12 months |
| 4 | Vitamin A supplement (200,000 IU) | Children 12 months to 59 months |

## Costing

An ingredient-based costing approach [15,16] using a mix of primary and secondary data was employed in this study, and the data were collected between October and December 2021. Program data was obtained from the campaign monitoring and evaluation records.

The methodological guidelines of Boonstoppel et al (2021) on how to cost immunization campaigns were adopted [17]. An independent public health specialist with experience in economic evaluation led the development of the cost analysis protocol, with input from various stakeholders following a series of virtual and physical meetings. A one-day workshop to clarify the scope, methodology, as well as the data extraction plan for the study was held with staff of Malaria Consortium Bauchi State field office (Monitoring and Evaluation Officer and Finance Officer), the Malaria Consortium Country Officers (Public Health Specialist, Finance Officer, Monitoring and Evaluation Officer, and other support staff), representatives from the Bauchi State Ministry of Health, and representative of Vitamin Angels (a partner non-governmental organization, which donated all the Vitamin A distributed during the study). This study team developed a costing template used to collect costs data from the state and country offices by the finance officers at state and country level. Cost data and other relevant data were also extracted from the programmatic, financial, and coverage data from monitoring and evaluation records. Program staff clarified the levels of staff effort to estimate the economic value of human resources input in the study, as SMC–VAS integration was an additional responsibility to their routine work. Microsoft Excel 2016 was used to aggregate all the extracted costs.

Data collection started in October 2021 after Cycle 1 had been completed, so only retrospective data were collected for this cycle. Data for Cycle 4 was collected prospectively during the intervention period and retrospectively afterwards to fill any missing gaps in the data collected. The time horizon for this study was one year, with 2021 as the baseline. The costs were aggregated to enable comparison between SMC with and without VAS administration. Due consideration was made for costs incurred by staff at the national malaria program and the Ministry of Health levels, by the Malaria Consortium, and by donors to the SMC-VAS project. There was no direct cost to the beneficiaries in this study, as Vitamin A and SPAQ were given at no tangible cost to the children during door-to-door campaigns. Hence the cost analysis did not include that of the beneficiaries.

## Cost categories

In this cost analysis, both economic and financial costs for delivery of the integrated services were assessed. Financial costs are typically captured in project financial records while economic costs were estimated based on assumptions, such as level of effort of human resources and market prices of relevant items [18]. We estimated economic costs in this analysis as the value of all resources utilized, regardless of the source of financing. The economic costs estimated also included the opportunity cost of labour as well as donations to the project (e.g., Vitamin A donated by Vitamin Angels). It was particularly important to outline the opportunity cost of labour as the labour cost captured in the financial records was for staff covering the 10 LGAs of the SMC campaign while this cost analysis covered only the two LGAs where the SMC-VAS integration campaign was being carried out [15].

The cost data were initially captured as capital and operating costs and originating from the MC field office in Bauchi State, and the MC country office. Operating costs are the costs of inputs used frequently and are used up in one year. Also, for comparability with other studies, campaign specific costs were delineated from shared costs. Campaign costs refer to those costs utilized solely for this integrated intervention, while shared costs are those costs that the integrated intervention shared with other ongoing interventions' program activities. Training of CDDs and supervisors occurred before the onset of Cycle 1, and additional training was held in Cycle 4 for the integration of Vitamin A into the SPAQ campaign.

Costs were fit into the following activity categories: distribution cost, drug cost (SPAQ and vitamin A separately), training, supplies, meetings, labor cost, supervision, and social mobilization. The subcategories under each of these cost categories (shown in Annex 1) were used in a cost-effectiveness study of a district-wide SMC intervention in Mali [16]. However, in this present study, the transportation cost/allowance was retained under the respective cost categories in which they occurred, as the drug distributors and supervisors where separately given transportation allowance. The cost of Vitamin A and SPAQ was estimated as the market price per dose of either drug (Table 3).

Fig 2 (Materials used for Cycle 1 (SMC) and 4 (SMC-VAS) campaigns, in Giade and Katagum LGAs, Bauchi State 2021) below shows some of the materials used for the SMC-VAS campaign.

Since the drug delivery in the integrated intervention was community based, the costs for actual delivery to the children were captured from the project financial records as the wages and benefits for the CDD and their supervisors, as well as other program related costs. No volunteer time was reported in this study. Costs not included in this analysis are:

1. Management costs of Malaria Consortium

2. Capital costs (vehicles, motorcycles, or health facilities)

3. Non-financial costs (e.g., volunteer time and recipient/family costs)

4. Coverage surveys and drug resistance monitoring

We excluded inventory and utility costs based on the recommendation from methodological guidance on costing immunization campaigns which assumes that the share of the building space allocated to a campaign of a limited number of days is small [15]. The cost is reported in Naira with conversion to USD at 2021-dollar value (NGN409.6 to USD1) based

**Table 3. Unit cost of SPAQ and Vitamin A used in the SMC-VAS integration intervention.**

| Commodities | Unit price/capsule (USD) in MONTH YEAR |
| --- | --- |
| Vitamin A 100,000 IU | 0.020 |
| Vitamin A 200,000 IU | 0.022 |
| SPAQ1 | 0.26 |
| SPAQ2 | 0.29 |

| Materials for SMC | Materials for VAS |
| --- | --- |
| <ul><li>**SPAQ blister packs for infant dose -** enough for all children 3-11 months of age</li><li>**SPAQ blister packs for child dose -** enough for all children 12-59 months of age</li><li>**Clean spoon and cup**</li><li>**Clean water**</li><li>**SMC Record Card -** to give to caregiver if the child has not been issued one in previous cycles</li></ul> | <ul><li>**Vitamin A 100,000 IU capsules -** enough for all children 6-11 months of age</li><li>**Vitamin A 200,000 IU capsules -** enough for all children 12-59 months of age</li><li>**Scissors -** to cut off the narrow tip of the vitamin A capsule</li><li>**Plastic Bag and bin-** to collect and dispose of used capsules</li><li>**Vitamin A stickers -** to record VAS and place on Child Health Card</li><li>**Grouwth Monitoring Register –** to record details of children given VAS</li></ul> |
| <ul><li>**Hand hygiene materials -** alcohol-based hand sanitizer or soap and clean water, clean towels</li><li>**SMC+VAS Job Aid -** to follow steps for administration of SPAQ and vitamin A</li><li>**SMC+VAS Tally Sheet -** to record the number of age appropriate doses distributed and the number of children reached with SPAQ and vitamin A</li><li>**Referral Form -** for children or who with danger signs, have fever, or who have severe respiratory illness or have difficulty breathing</li><li>**Pens -** for recordkeeping purposes</li><li>**Chalk –** for marking houses</li><li>**Map -** for daily activity</li></ul> ||

**Fig 2. Materials used for Cycle 1 (SMC) and 4 (SMC-VAS) campaigns, in Giade and Katagum LGAs, Bauchi State 2021.**

on the average (median) Naira-Dollar exchange for year 2021 as provided by the Central Bank of Nigeria. The percentage share of associated cost categories of the total cost was also estimated.

Total cost was estimated first for SMC only and then SMC-VAS integrated intervention. The cost per child receiving only SMC, and receiving both SMC and VAS were also estimated. This was straightforward in Cycle 1 as the children only received SPAQ. In Cycle 4, consideration was made for both those children who received SMC alone and SMC-VAS integration because the age range for SMC was broader than for VAS.

## Sensitivity analysis

Univariate sensitivity analysis was conducted to evaluate how specific cost components affected the cost per child receiving only SMC in the standalone SMC cycle, either SMC alone or both Vitamin A and SMC in the integration cycle, and only SMC in the integration cycle. We examined four cost components: distribution, labor, SPAQ, and supplies, and applied a 10% increase and decrease to each to allow for a robust assessment of how changes in these cost factors might affect the overall cost of the intervention. For each scenario, we calculated the resultant 'cost per child' and quantified its sensitivity by measuring deviations from the baseline. These deviations were then represented visually using 'tornado plots'.

Research ethics approval for the study was received from the Bauchi State Ethics Review Committee and National Health Research and Ethics Committee.

## Results

The cost of delivering SMC as standalone and the cost of adding VAS to an existing SMC program. The total cost for SMC only was $158,934 while for SMC-VAS it was $186,426 (Table 4). The largest driver of cost for both types of campaigns was the cost of distribution accounting for 30.6% and 31.9% for SMC and SMC-VAS respectively. This was followed by

**Table 4. The main cost categories of cost analysis, and proportion of total cost, for the SMC campaign in Cycle 1 (SMC) and 4 (SMC-VAS) campaigns, in Giade and Katagum LGAs, Bauchi State 2021.**

| Cost category | SMC | | SMC-VAS | |
|---|---|---|---|---|
| | Cost ($) | Proportion | Cost ($) | Proportion |
| Distribution cost | 48,565.67 | 30.6 | 59,472.34 | 31.9 |
| SPAQ | 47,771.74 | 30.1 | 48,213.71 | 25.9 |
| Training | 28,783.33 | 18.1 | 45,544.69 | 24.4 |
| Supplies | 25,114.65 | 15.8 | 13,748.13 | 7.4 |
| Meetings | 1,958.13 | 1.2 | 7,523.41 | 4.0 |
| Labour cost | 6,301.28 | 4.8 | 6,746.47 | 3.6 |
| Supervision | 439.42 | 0.3 | 945.60 | 0.5 |
| Vit A | – | – | 3,416.90 | 1.8 |
| Social mobilization | – | – | 815.36 | 0.4 |
| Total | **158,934.21** | **100.0** | **186,426.60** | **100.0** |

the cost of SPAQ, which accounted for 30.1% and 25.9% respectively. The proportion of the cost of training was higher for SMC-VAS (24.4%) compared with SMC (18.1%). The cost of holding meetings (which covers expenses such as venue costs, transportation allowances, lunch, accommodation, and per diem, particularly for participants from out of station) was much higher for SMC-VAS at $7,523 compared to SMC at $1,958. This cost difference of $5565 accrued from additional meetings that needed to be held with stakeholders for the introduction of Vitamin A into the SMC campaign cycle. The cost of supplies for SMC was significantly higher at $25,114 than for SMC-VAS at $13,748. Similarly, supplies took a higher proportion of the cost for SMC (15.8%) compared with SMC-VAS (7.4%). This was because many of the items procured at Cycle 1 were planned for the entire SMC round and items such as visibility materials and job aids were also reused in Cycle 4.

The cost of supervision and distribution was higher for SMC-VAS compared to SMC because distribution occurred for five days for SMC-VAS, and for four days for SMC. Overall, although the cost of the Vitamin A distributed was only $3,440, integrating Vitamin A into Cycle 4 added $27,492 over the cost of Cycle 1. Table 5 shows the breakdown of the main-cot categories by financial and opportunity cost categories.

**Table 5. The main cost categories of cost analysis by financial and opportunity costs for the SMC campaign in Cycle 1 (SMC) and 4 (SMC-VAS) campaigns, in Giade and Katagum LGAs, Bauchi State 2021.**

| Cost Categories | SMC | | SMC-VAS | |
|---|---|---|---|---|
| | Financial Cost ($) | Opportunity cost ($) | Financial Cost ($) | Opportunity cost ($) |
| Distribution cost | 48,565.67 | | 59,472.34 | |
| SPAQ | 47,771.74 | | 48,213.71 | |
| Supervision | 439.42 | | 945.60 | |
| Supplies | 25,114.65 | | 13,748.13 | |
| Training | 28,783.33 | | 45,544.69 | |
| Meetings | 1,765.88 | 192.24 | 7,523.41 | |
| Social mobilization | | | 815.36 | |
| Labor cost | | 6,301.28 | | 6,746.47 |
| Vitamin A | | | | 3,416.90 |
| Total | **159,469.49** | **6,493.53** | **176,263.23** | **10,163.37** |

 

Table 6 shows the coverage of SMC in Cycle 1 and SMC-VAS in Cycle 4. A total of 168,820 children under 5 years received SPAQ1 and SPAQ2 in the two study LGA in Cycle 1 while it was 170,681 children in Cycle 4. Among the children that received SPAQ1 or SPAQ2 in Cycle 4, 158,258 also received Vitamin A supplements during the campaign. Table 7 shows the cost per child receiving only SMC in the standalone SMC cycle, either SMC alone or both Vitamin A and SMC in the integration cycle, and only SMC in the integration cycle. For children who received only SMC in Cycle 1, we estimate the economic cost to be $0.94 per child, while it cost $1.18 per child to provide eligible children with combination of SPAQ and Vitamin A during an SMC cycle.

In the sensitivity analyses (Annex 2), the 'SMC only' cycle showed a $0.03 sensitivity on cost per child to changes in distribution costs. The other cycles, SMC±VAS and SMC+VAS, demonstrated a similar sensitivity of about $0.04. In contrast, the 'Supplies' component had minimal impact on the 'cost per child' across all cycles, indicating its low sensitivity. The effects of 'Labour cost' and 'SPAQ cost' components varied, affecting the 'cost per child' by about $0.01 to $0.03. The implementers had adjusted the operational strategy by instituting a 30-minute wait between the administration of SMC medicines and VAS, and by revising the daily distribution targets from 70 to 56 children, with an added implementation day. These adjustments, aimed at maintaining the quality of the campaign without significantly elevating costs, are supported by the sensitivity analysis. This analysis validates the implementers' decisions, suggesting that the operational modifications aligned with the empirical cost sensitivities identified post-implementation.

## Discussions

The findings from this study shed light on the intricacies of incorporating Vitamin A supplements into SMC campaigns, highlighting both the associated costs and the primary cost drivers. The analysis reveals that the cost to cover a child for one SMC cycle is less than $1.00, suggesting that the expense for all four cycles would not exceed $4.00 per child, as subsequent cycles do not incur the initial set-up costs. These estimates are consistent with previous research; for example, Baba et al. (2020) reported an estimated weighted average economic cost of administering four cycles at $2.71 to

**Table 6. Coverage and quantity of drugs SPAQ and Vitamin A capusles used in Giade and Katagum LGAs, Bauchi State in Cycle 1 and 4 of the SMC and SMC-VAS campaigns, 2021.**

| Coverage category | Cycle 1 | | | Cycle 4 | | |
|---|---|---|---|---|---|---|
| | 3 to <12m | 12 to <59m | Total | 3 to <12m | 12 to <59m | Total |
| **SPAQ** | | | | | | |
| Number of children who received SPAQ | 32769 | 136051 | **168820** | 33539 | 137142 | **170681** |
| Number of blisters wasted* | 157 | 301 | **458** | 58 | 143 | **201** |
| Total Number of SPAQ capsules used | **32926** | **136352** | **169278** | **33597** | **137285** | **170882** |
| **Vitamin A** | | | | | | |
| Number of children who received Vitamin A | | | | 20498 | 137378 | **157876** |
| Number of capsules wasted* | | | | 99 | 243 | **342** |
| Total Number of Vitamin A capsules used in cycle | | | | **20547** | **137711** | **158258** |

*Each blister/capsule represents *a* child's dose.

**Table 7. Estimation of Cost per child receiving only SMC (Cycle 1), both Vitamin A and SMC, and only SMC (Cycle 4), only both Vitamin A and SMC, or only SMC (Cycle 4) in the study LGAs.**

| Cycle | Offered to Child | Cost Indicator: Total Cost… | Total Cost ($) | No. of Children reached | Cost per Child ($) |
|---|---|---|---|---|---|
| 1 | SMC alone | per child receiving only SMC | 158,934.22 | 168820 | 0.94 |
| 2 | SMC+VAS | per child receiving both Vitamin A and SMC | 186,426.61 | 157876 | 1.18 |

$8.20 per child across various African countries utilizing a door-to-door distribution strategy [15]. Conversely, a study in Ghana estimated a range from $4.61 to $26.14 per child, influenced by regional differences [19]. Moreover, a comprehensive delivery assessment in Senegal found economic costs per SMC course ranging from $0.38 to $2.74 [5]. These comparative findings emphasize the importance of contextualized strategies for implementing SMC programs and call for a more rigorous comparative analysis to account for differences in methodology, regional economic conditions, and implementation scale. However, it is important to point out the methodology for calculating costs differed among these studies.

Additionally, the study indicated that incorporating Vitamin A into a standard SMC cycle led to an incremental cost of just $0.24 per child covered for both SMC and VAS. When considering the broader scope of the campaign, which encompasses all eligible children, the incremental cost further decreased to $0.11 per child. This has profound implications for the economic viability of adding Vitamin A supplementation to SMC programs. The nominal additional cost implies that it could be an attractive policy for public health officials, offering a dual benefit of malaria prevention and enhanced nutritional status for at-risk, underserved populations. Substantial benefits that may arise from the modest additional per-child cost, including the potential to reduce both malaria morbidity and mortality and to improve Vitamin A status. A study by Shankar et al. (1999) among children under five in Papua New Guinea indicated that VAS is an economical approach to decrease morbidity caused by P. falciparum, with a 30% reduction in illness episodes among supplemented children compared to those without supplements [20,21]. Thus, incorporating Vitamin A into SMC, a low-tech yet effective intervention, appears to be a cost-effective strategy for advancing child health.

Detailed examination of the integration costs reveals that, although initial expenses for training and material procurement are one-off, other costs such as monitoring and supply chain management could recur with each cycle, necessitating a nuanced understanding of this cost structure for sustained integration. The investment in training and meetings necessary to integrate VAS into SMC represents a higher initial cost that should diminish over time once the integration process becomes routine. These upfront investments in capacity building are expected to result in long-term enhancements in both the efficiency and efficacy of the program. As the teams involved gain proficiency in delivering combined services, the resources needed for subsequent initial and refresher trainings and meetings are projected to decrease, potentially lowering the overall cost per child in future cycles and bolstering the sustainability of the integrated approach. This anticipated decrease in training and meeting costs aligns with the need for accurate data and distribution cost estimates, as highlighted by the sensitivity analysis which underscores the 'distribution' component as a critical cost driver.

Similarly, materials acquired for the SMC campaign, such as educational resources, medication packages, and health worker protective equipment, can be used throughout the four SMC cycles. Given that VAS is integrated into only one of the four SMC cycles annually, the costs for these materials can be classified as initial expenditures. This study accounted for these costs only in the context of the standalone SMC to reflect the integration pattern within the four-cycle SMC framework. Spreading these costs over multiple cycles could further highlight the cost-effectiveness of the integrated model.

## Study limitations

This study's cost analysis, limited to two cycles, one with Vitamin A integration and one without, does not provide a comprehensive estimate of the cost per child over the full four cycles of SMC or for exclusive Vitamin A supplementation (VAS). Also, the timing of integration—whether during the 4th cycle with VAS/SMC or the 1st cycle, when engagement in SMC is potentially higher—may influence outcomes. Moreover, the analysis does not account for the expenses incurred by the beneficiaries' parents or guardians participating in the stand-alone or integrated campaign. This gap is due to the door-to-door drug delivery strategy employed in this study, which excludes the opportunity cost related to the time spent by parents or guardians with the drug distributors. Additionally, incorporating administrative costs beyond the country office of the implementing organization (i.e., Malaria Consortium) in the costs would notably increase the total campaign

cost. Nevertheless, this study conformed to established costing guidelines for such campaigns by excluding these above-country overhead costs, thereby ensuring that its findings remain consistent and relevant to similar public health initiatives [15]. While some elements of the cost analysis, like 'supplies,' confirm the soundness of our methodology, as its variations minimally impact the overall result, others, specifically 'distribution,' indicate areas of potential risk to the success of integration programs. Nonetheless, it is crucial to note the study was a cost analysis that did not assess the cost-effectiveness in achieving specific health outcomes, including decreased under-5 mortality, lower malaria incidence among under-5s, and reductions in malnutrition rates.

In this study, cost analyses were performed for children receiving only Seasonal Malaria Chemoprevention (SMC) and those receiving both SMC and Vitamin A Supplementation (VAS). The distinction was straightforward in first cycle, where only SPAQ was administered. In contrast, Cycle 4 required nuanced consideration of two distinct categories—children receiving only SMC and those part of the integrated SMC-VAS program—attributable to the broader age range eligible for SMC compared to VAS. While this approach is critical to ensure that interventions are comprehensive and meet the varied needs of the target population, future studies may refine their assessment of the integration strategy to focus exclusively on the age groups eligible for each specific intervention. Finally, the applicability of this analysis across different Nigerian settings may be limited due to variability in supply and distribution costs.

## Conclusion

The cost per child for an SMC standalone campaign and an SMC-VAS integrated campaign was estimated using an ingredient-costing methodology. This analysis of both financial and economic costs provides insights into the feasibility of integrating Vitamin A distribution into the standard SMC cycle, revealing an additional cost of $0.24 per child. This incremental cost is relatively minimal, suggesting that the integration of Vitamin A delivery into an SMC cycle is a worthwhile investment, especially considering the health benefits for children under five years of age. Although scaling up a routine SMC-VAS program is advisable, future research should focus on conducting a cost-effectiveness analysis of their integration, incorporating population-based coverage surveys and reported rates of malaria, nutritional deficiencies, and overall child morbidity and mortality.

## Supporting information

**S1 Data. Data set and cost categories.**
(XLSX)

**S2 Data. Annex 1: Main cost categories and associated cost subcategories.**
(DOCX)

**S3 Data. Annex 2: Sensitivity analyses.**
(DOCX)

**S4 Data. Inclusivity questionnaire.**
(PDF)

## Acknowledgments

The authors acknowledge Dr. Perpetua Uhomoibhi, the former National Coordinator, NMEP and Dr. Nnenna Ogbulafor, the Head of Case management branch, NMEP, for their support in stakeholder engagement. We appreciate all members of the Malaria Consortium country office Finance staff, Bauchi state project team, the Executive Chairman, Bauchi State Agency for the Control of AIDS, Tuberculosis and Malaria, the Executive Secretary, Bauchi state Primary Health Care Board; the state Malaria Program Manager, Alhaji Babuga, and all others not mentioned but supported the data collection

and validation process. We express our appreciation to the community drug distributors and health facility workers, who support SMC delivery to eligible children in communities and participated in this study.

## Author contributions

**Conceptualization:** Olusola Bukola Oresanya, Abimbola Phillips, Emmanuel Shekarau, Eva S. Bazant.

**Data curation:** Olujide Arije, Jesujuwonlo Fadipe, Abimbola Phillips.

**Formal analysis:** Olujide Arije.

**Funding acquisition:** Olusola Bukola Oresanya.

**Methodology:** Olujide Arije, Abimbola Phillips.

**Project administration:** Olusola Bukola Oresanya, Kunle Rotimi, Kolawole Maxwell, Abimbola Phillips, Nneka Onwu.

**Resources:** Kolawole Maxwell, Eva S. Bazant.

**Supervision:** Olusola Bukola Oresanya, Abimbola Phillips.

**Validation:** Olusola Bukola Oresanya.

**Writing – original draft:** Abimbola Phillips.

**Writing – review & editing:** Olusola Bukola Oresanya, Olujide Arije, Jesujuwonlo Fadipe, Kunle Rotimi, Emmanuel Shekarau, Nneka Onwu, Eva S. Bazant.

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
