## [Decision Letter · Decision Letter 0]

30 Jan 2025

Dear Dr. Bukola

Thank you for submitting your manuscript to PLOS ONE. After careful consideration, we feel that it has merit but does not fully meet PLOS ONE’s publication criteria as it currently stands. Therefore, we invite you to submit a revised version of the manuscript that addresses the points raised during the review process.

**ACADEMIC EDITOR:**
publication criteria

Please submit your revised manuscript by February, 28 If you will need more time than this to complete your revisions, please reply to this message or contact the journal office at plosone@plos.org . A rebuttal letter that responds to each point raised by the academic editor and reviewer(s). You should upload this letter as a separate file labeled 'Response to Reviewers'.A marked-up copy of your manuscript that highlights changes made to the original version. You should upload this as a separate file labeled 'Revised Manuscript with Track Changes'.An unmarked version of your revised paper without tracked changes. You should upload this as a separate file labeled 'Manuscript'.

We look forward to receiving your revised manuscript.

Kind regards,

José Luiz Fernandes Vieira

Academic Editor

PLOS ONE

Journal Requirements:

“This work was supported, in whole or in part, by the Bill & Melinda Gates Foundation (Grant Number INV‐01076 to the Task Force for Global Health’s Health Campaign Effectiveness Program). Under the grant conditions of the Foundation, a Creative Commons Attribution 4.0 Generic License has already been assigned to the Author Accepted Manuscript version that might arise from this submission"

Reviewers' comments:

Reviewer's Responses to Questions

**Comments to the Author**

1. Is the manuscript technically sound, and do the data support the conclusions?

Reviewer #1: Partly

2. Has the statistical analysis been performed appropriately and rigorously?

Reviewer #1: N/A

3. Have the authors made all data underlying the findings in their manuscript fully available?

Reviewer #1: Yes

4. Is the manuscript presented in an intelligible fashion and written in standard English?

Reviewer #1: Yes

Reviewer #1: Dear authors

Thanks for allowing me to review this interesting work. Overall I think this is important work and worth publishing. However, I think the authors need to reconsider their methods and focus as currently presented. They have compared costs for two different locations and timepoints and for 1 cycle of a 4 or 5 cycle intervention - SMC. I feel like there could be many potential confounders and that this is a very serious limitation. While the authors point this out in the limitations I wonder if they considered broadening their analysis to cover all cycles in the intervention and/or could adjust for any confounders in location or delivery model and timings in the analysis and present this? They compare to other cost studies which show a total intervention cost also so it would be better to reflect the same in their analysis if possible.

Some other specific points:

Layout and structure - the whole piece needs to be reviewed and ensure the correct elements are in the correct sections - e.g. currently elements of the discussion around the inclusion and rationale for specific costs are included in the results - there need to be moved to the correct sections

Ethics - please add date and reference for approval

Readability - please check the text for readability overall - there are a few instances where it doesn't make sense - for example in L209 - this is not a sentence.

Formatting - the authors should check text formatting as the fonts etc are not uniform

Happy to review again once these comments are addressed.

**Do you want your identity to be public for this peer review?** For information about this choice, including consent withdrawal, please see our Privacy Policy

Reviewer #1: No

---

## [Author Response · Author response to Decision Letter 1]

19 Jun 2025

Reviewer's comment 1:

Overall, I think this is important work and worth publishing. However, I think the authors need to reconsider their methods and focus as currently presented.

1. They have compared costs for two different locations and timepoints and for 1 cycle of a 4 or 5 cycle intervention - SMC. I feel like there could be many potential confounders and that this is a very serious limitation. While the authors point this out in the limitations I wonder if they considered broadening their analysis to cover all cycles in the intervention and/or could adjust for any confounders in location or delivery model and timings in the analysis and present this? They compare to other cost studies which show a total intervention cost also so it would be better to reflect the same in their analysis if possible.

Response to comment 1:

We would like clarify that, in terms of locations, the cycles being compared for the cost analysis were carried out in the same locations.

Based on the intervention design, there were 4 cycles of SMC in the study location as at the time of this study. Cycles 1 – 3 were designed to run the same way with the delivery of SMC-only from house-to-house, while cycle 4 was designed to deliver vitamin A in addition to SMC from house-to-house. We don’t have any indication that this was not so, therefore we believe that taking cost estimates for one of the three previous cycles for comparison with the cycle where SMC-VAS was delivered is rational. However, to rule out any possible confounding, we included a sensitivity analyses in our result focusing explicitly on critical cost components (distribution, SPAQ, labour, and supplies). These analyses evaluated the robustness of our findings by systematically varying costs within plausible ranges (±10% to ±20%) and assessed their impact on total costs per child. The findings, detailed in Annex 2 of our submission documents, showed moderate sensitivity to changes in distribution costs ($0.03–0.04 per child), low-to-moderate sensitivity to labour and SPAQ costs ($0.01–0.03 per child), and minimal sensitivity to supplies costs.

Reviewer comment 2:

Some other specific points:

Layout and structure - the whole piece needs to be reviewed and ensure the correct elements are in the correct sections - e.g. currently elements of the discussion around the inclusion and rationale for specific costs are included in the results - there need to be moved to the correct sections.

Response to comment 2:

We have revised accordingly in tracked changes and moved discussion elements to the discussion section, where we have provided additional contextual information and interpretation to enhance clarity.

Reviewer's comment 3:

Ethics - please add date and reference for approval

Response to comment 3:

We have revised the ethical consideration section accordingly to reflect the data and approval reference number.

Reviewer's comment 4:

Readability - please check the text for readability overall - there are a few instances where it doesn't make sense - for example in L209 - this is not a sentence.

Formatting - the authors should check text formatting as the fonts etc are not uniform

Response to comment 4:

We apologize for the typographical errors in the document. We have revised accordingly.

Thank you for your useful feedback.

---

## [Decision Letter · Decision Letter 1]

1 Aug 2025

A cost analysis comparing Seasonal Malaria Chemoprevention with and without Vitamin A Supplementation among under-5 children in Nigeria

PONE-D-24-52126R1

Dear Dr. Olusola

We’re pleased to inform you that your manuscript has been judged scientifically suitable for publication and will be formally accepted for publication once it meets all outstanding technical requirements. The reviewer accepts all changes made by the authors and recommends the manuscript for publication.Within one week, you’ll receive an e-mail detailing the required amendments. When these have been addressed, you’ll receive a formal acceptance letter and your manuscript will be scheduled for publication.

Kind regards,

José Luiz Fernandes Vieira

Academic Editor

PLOS ONE

Reviewers' comments:

Reviewer's Responses to Questions

**Comments to the Author**

Reviewer #1: All comments have been addressed

2. Is the manuscript technically sound, and do the data support the conclusions?

Reviewer #1: Yes

3. Has the statistical analysis been performed appropriately and rigorously?

Reviewer #1: Yes

4. Have the authors made all data underlying the findings in their manuscript fully available?

Reviewer #1: Yes

5. Is the manuscript presented in an intelligible fashion and written in standard English?

Reviewer #1: Yes

Reviewer #1: Thanks - all comments addressed and article looks good now. I don't have any further comments at this time and happy for this publications to now be published.

**Do you want your identity to be public for this peer review?** For information about this choice, including consent withdrawal, please see our Privacy Policy

Reviewer #1: No

---

## [Editor Report · Acceptance letter]

PONE-D-24-52126R1

PLOS ONE

Dear Dr. Oresanya,

I'm pleased to inform you that your manuscript has been deemed suitable for publication in PLOS ONE. Congratulations! Your manuscript is now being handed over to our production team.

Kind regards,

on behalf of

Dr. José Luiz Fernandes Vieira

Academic Editor

PLOS ONE